# The effect of angiotensin receptor blockers on peripheral artery disease progression: A retrospective cohort study in Jordan

Mahar Zaid Kasasbeh[1]*, Eilaf Mohammad Al-Rabab'ah[1], Rawand A. Khasawneh[1], Fadi N. Asfar[2], Samah F. Al-Shatnawi[1], Nour A. Al-Sawalha[1]

1 Department of Clinical Pharmacy, Faculty of Pharmacy, Jordan University of Science and Technology, Irbid, Jordan, 2 Center of Pharmacometrics and System Pharmacology, Department of Pharmaceutics, College of Pharmacy, University of Florida, Florida, United States of America

* mzkasasbeh17@ph.just.edu.jo

## Abstract

### Background

Peripheral artery disease (PAD) is a common atherosclerotic vascular condition associated with many complications. The specific effect of angiotensin receptor blockers (ARBs) on the short-term complications in PAD patients has not been explored before.

### Objectives

This study aimed to evaluate the association between ARBs use and the disease severity, progression, the frequency of surgical interventions, and arterial occlusion rates among users.

### Methods

A retrospective cohort study was conducted at King Abdullah II Hospital, including 133 patients followed between 2019 and 2024. The cohort was divided into two groups based on their exposure to ARBs: the ARBs group and the control group. Data related to patient demographics, clinical characteristics, medical history, laboratory results, vital signs, in addition to clinical outcomes were extracted from electronic medical records. Statistical analysis was performed using SPSS version 27.0.

### Results

Among the participants, 24.8% experienced amputation, with 21.1% in the control group compared to 3.8% in the ARBs group. ARBs group has 79.2% lower Limb amputation (P-value 0.011; OR= 0.208, 95% CI 0.062–0.700) and 71% lower likelihood of having advanced PAD stage (stages III and IV) (P-value = 0.034, OR= 0.294,

**Data availability statement:** All relevant data are within the paper and its Supporting Information files.

**Funding:** The author(s) received no specific funding for this work.

**Competing interests:** The authors have declared that no competing interests exist.

95% CI = 0.095–0.911) compared to the control group. Linear regression analysis showed that the ARBs group has a decreased frequency of revascularization procedures (P-value 0.025;95% CI −1.004 − - 0.068).

## Conclusion

ARBs may serve as a promising protective treatment option for PAD patients, potentially benefiting disease progression by lowering the risk of amputation and reducing the need for revascularization procedures.

## Introduction

Peripheral Artery Disease (PAD) is a chronic atherosclerotic vascular condition characterized by the narrowing or occlusion of arteries, most commonly in the lower limbs, which reduces the blood supply and causes ischemic symptoms. It is a widespread condition affecting over 200 million people worldwide, with an estimated prevalence of 6% among adults [1]. It is the third leading cause of atherosclerotic cardiovascular morbidity and mortality, following coronary artery disease and stroke [2]. The prevalence of both asymptomatic and symptomatic PAD overlaps significantly with coronary artery disease and cerebrovascular diseases due to their shared atherosclerotic pathogenesis. Accordingly, PAD can be used to pinpoint subjects at high risk for cardiovascular events. Given this substantial overlap, the ankle-brachial index – a standard non-invasive diagnostic and screening tool for PAD – should be routinely measured during clinic visits, especially in patients with established cerebrovascular diseases, coronary artery disease, or those at high risk. Routine ankle-brachial index measurements can help mitigate the frequent underdiagnosis of peripheral artery disease (PAD) and facilitate timely interventions, thereby improving clinical outcomes. [3] Several established atherosclerotic risk factors contribute to the development of PAD, with diabetes mellitus (DM), hypertension (HTN), dyslipidemia, and smoking being among the most significant [1,4]. PAD progresses from an initially asymptomatic condition to a chronic symptomatic disease, characterized by ischemic-related exertional symptoms such as intermittent claudication. In advanced stages, it can lead to critical limb ischemia and, in severe cases, result in limb amputation.

Current treatment approaches for PAD include lifestyle modifications – especially smoking and alcohol cessation, and weight management. Secondary atherosclerotic prophylaxis medications such as antiplatelets (primarily clopidogrel), antithrombotic therapy in patients with low bleeding risk (low-dose rivaroxaban combined with aspirin), and statins. Managing comorbid conditions is crucial, including DM, dyslipidemia – with an aggressive low-density Lipoprotein goal of < 70 mg/dl and 50% reduction from baseline – and HTN with a goal blood pressure of < 130/80 mmHg. For symptomatic management, Supervised exercise therapy is now considered a first-line treatment. Additionally, Vasodilators such as cilostazol are traditionally used to improve walking distance and claudication. In advanced stages, surgical revascularization interventions may be necessary [5,6].

Current guidelines recommend Angiotensin Converting Enzyme Inhibitors (ACEIs) or Angiotensin Receptor Blockers (ARBs) for individuals with PAD and HTN to reduce major adverse cardiac events (MACE) [5]. Both MACE and major adverse limb events (MALE) are considered the major complications of PAD and have been associated with increased mortality [7]. While prior studies have linked ACEIs and ARBs to reduced MACE and MALE rates, their direct role in modifying PAD severity and progression, arterial occlusion, and the need for revascularization has not been systematically examined [7–13]. Progression in PAD can be assessed by worsening disease classification, increased need for surgical interventions, and greater arterial occlusion severity detected via imaging techniques [2,14–16].

While both ACEIs and ARBs have demonstrated positive clinical effects on PAD, our hypothesis was based on that, unlike ACEIs, ARBs' effects on endothelial function and vascular inflammation are independent from blood pressure reduction [15]. This study specifically focuses on the utilization of ARBs among patients with PAD in Jordan, aiming to investigate their impact on disease severity and progression, the frequency of surgical interventions, and arterial occlusion rates as identified by angioplasty or Computed Tomography Angiography (CTA). By addressing this gap, the study will provide insights into the potential role of ARBs in PAD management beyond their cardiovascular benefits, informing future clinical strategies.

## Methods

To investigate the direct effect of ARBs on the short-term complications of PAD, we conducted a retrospective, observational cohort analysis study in the Surgery Department of King Abdullah II University Hospital (KAUH) in Jordan, utilizing electronic health records from January 2024 to January 2025. Ethical approval was secured the Institutional Review Board (IRB approval number 2023/585). Patient confidentiality was preserved through the de-identification of data during both the collection and analysis phases.

The study cohort comprised adult patients (aged 18 years and older) who visited the surgery department ward or clinic between 2019 and 2024 and were diagnosed with PAD via CTA. Patients were identified from the hospital's electronic health records using the International Classification of Diseases, 10th Revision codes (ICD-10) for the following conditions: PAD, angioplasty, CTA, and lower limb ischemia. Individuals who were taking ACEIs or had discontinued these medications within the last two years, as well as those with incomplete records, were excluded from the study. The cohort was then divided into two groups based on their exposure to ARBs: those who had been on ARBs for at least two years (exposed group) and those who had not (unexposed group). Well-documented notes from physicians or pharmacists were utilized to confirm the exposure status and mitigate the risk of mis-classification bias.

Data were extracted from the hospital's electronic medical records and included various patient demographics such as age and sex, clinical characteristics including smoking status and current medications, and medical history covering conditions like HTN, DM, dyslipidemia, and the date of PAD diagnosis. Additionally, the data encompassed laboratory results and vital signs, including a lipid panel, HbA1c, and blood pressure measurements, as well as clinical outcomes, which comprised the number of angioplasties the patient underwent, limb status, PAD stage, and findings from the most recent angioplasty.

Regarding the staging system, the Fontaine classification was used. Accordingly, patients were classified to one of the following: stage I (Asymptomatic, incomplete blood vessel obstruction), stage II (Mild claudication pain in limb), stage IIA (Claudication at a distance > 200 m), stage IIB (Claudication at a distance < 200 m), stage III (Rest pain, mostly in the feet), stage IV (Necrosis and/or gangrene of the limb) [17].

1    To mitigate investigator bias, the eligibility assessment for participants was conducted separately from the data collection process. Furthermore, the investigator responsible for data collection utilized a distinct Excel sheet containing only the Identification (ID) numbers of eligible patients, devoid of any information regarding exposure status. The data collection began with an investigation of clinical outcomes occurring from the date of recruitment, looking back over the previous five years, followed by the collection of laboratory results and vital signs, then moving to medical history, clinical characteristics, and finally, demographics. Notably, no data were missing.

Data were analyzed using the Statistical Package for the Social Sciences (SPSS) version 27.0. Continuous variables were reported as means ± standard deviations (SD), and variances between the two groups were assessed with Levene's test. Categorical variables were expressed as frequency, percent, and mode, and differences between groups were evaluated using the Chi-square test. Multivariate regression analysis was conducted to examine the association between ARBs and various outcomes, adjusting for the following confounding factors: duration of diagnosis, HTN, DM, dyslipidemia, smoking status, blood pressure category (where SBP < 130 mmHg is considered controlled) and HbA1c levels category (where HbA1c < 7% is considered controlled).

The type of regression analysis employed was based on the nature of the outcome: linear regression for continuous outcomes, and logistic regression for binary outcomes. The Chi-square test was also employed to evaluate the relationship between exposure and categorical variables (e.g., antiplatelet usage and statin intensity). Statistical significance was defined as two-sided with a P-value < 0.05.

The existing literature primarily provides data regarding the hazard ratio (HR) of amputation between exposed and unexposed groups in patients diagnosed with peripheral artery disease. Consequently, it was not feasible to directly utilize an odds ratio (OR) derived from the literature for calculating the minimum sample size required for our study. However, it is recognized that HR may approximate OR in instances of rare events, defined as those with a probability of occurrence less than 10%. The EUCLID trial reports an overall major amputation rate of 1.6%, with specific rates of 1.2% in patients with non-critical limb ischemia and 8.4% in patients with critical limb ischemia [18]. Our study cohort reflects a similar distribution of patients with non-critical limb ischemia and critical limb ischemia, leading to the assumption that major amputation events would be infrequent within our sample. Accordingly, we referenced the major amputation HR of 0.89 as reported by N. Elsayed and colleagues to estimate the sample size necessary for our research [13]. Utilizing G*power software, we established the following parameters: OR = 0.89, two-tailed alpha error = 0.05, and power = 0.80, which resulted in a calculated total sample size of 3,621. However, due to the practical nature of our study, the stringent inclusion criteria for the defined cohorts, and the limitation of recruiting from a single tertiary hospital, we were able to enroll only 133 patients, maintaining an unexposed-to-exposed ratio of 3:1. The sampling process is depicted in the accompanying figure (Fig 1). A post hoc power analysis was performed using G*power software, incorporating the observed odds ratio of amputation, the actual sample size, and the two-tailed alpha error of 0.05, which yielded a calculated power of 0.99.

## Results

The study comprised 133 PAD patients identified from the databases of KAUH between January 2024 and January 2025. Within this cohort, 36 patients were assigned to the ARBs group, while 97 patients were placed in the control group. The majority of participants were male (92.5%), with ages spanning from 36 to 85 years, and a mean age of 61.1 ± 10.32 years. Baseline characteristics were comparable between the ARBs and control groups, with the exception of a few variables: HTN status, blood pressure category, and HbA1C category. These variables were accounted for in the analyses. Table 1 presents the baseline clinical characteristics of the study population, highlighting the differences between individuals in the ARBs group and those in the control group. Notably, 24.8% of the sample experienced amputation, with 21.1% in the control group compared to only 3.8% in the ARBs group.

In the ARBs group, the absolute amputation rate was 13.8% (n = 5) of patients had an amputated limb, compared to 28.9% (n = 28) in the control group. Logistic regression was used to compare the odds of amputation between the two groups. The odds of amputation in the ARBs group was 79.2% lower than the control group (P-value 0.011; OR = 0.208, 95% CI 0.062–0.700), adjusted for smoking status, DM, HTN, dyslipidemia, duration of diagnosis, blood pressure category (controlled vs. uncontrolled), and HbA1c category (controlled vs. uncontrolled).

The frequency of surgical interventions performed on patients from 2019 to 2025 was compared between groups using linear regression. In the ARBs group, the β coefficient was −0.536 (P-value 0.025; 95% CI −1.004 – - 0.068), adjusted for the same variables mentioned above. Additionally, using the same model, the number of occluded arteries found during

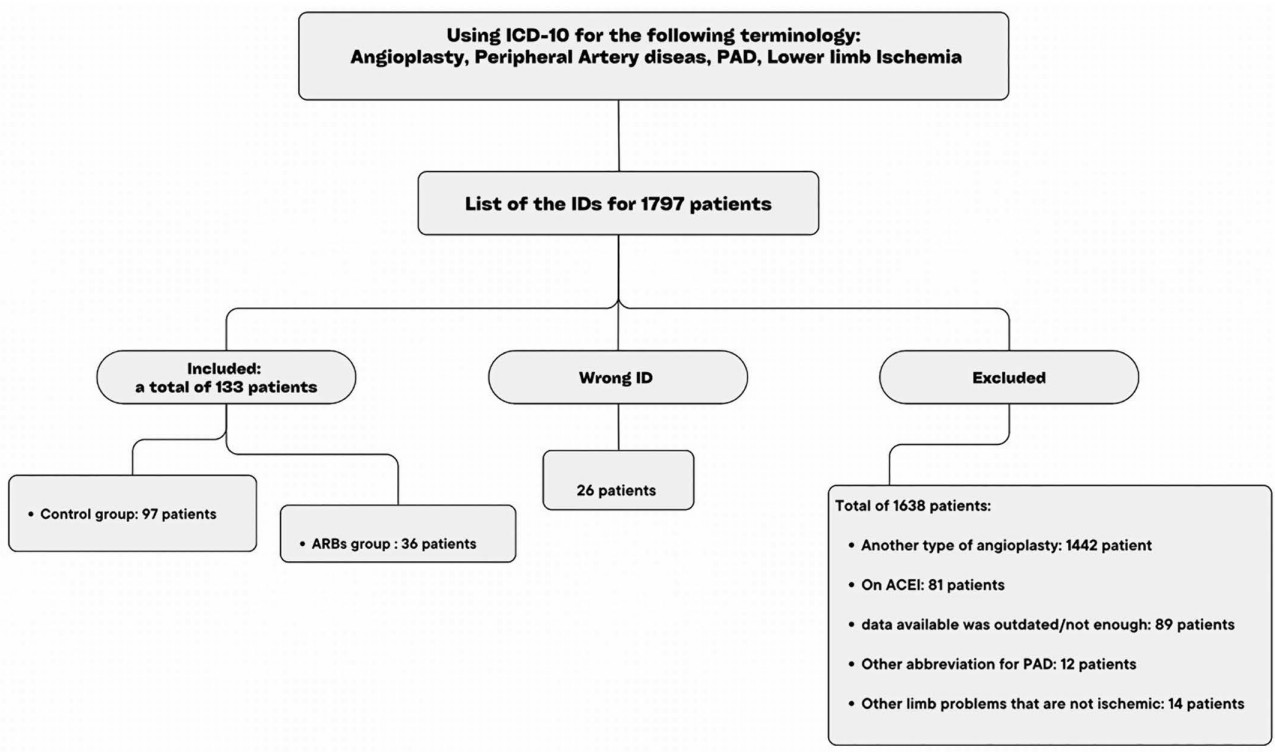

**Fig 1. Sampling flowchart.**

the most recent surgical intervention was also compared between groups, revealing a β coefficient of 0.019 for the ARBs group (P-value 0.965, 95% CI −0.849–0.888).

Accordingly, the odds of having more advanced disease stages between the two groups were compared using logistic regression. The OR of having a more advanced disease stage in the ARBs group was 0.294 (P-value 0.034, 95%CI 0.095–0.911) adjusted for smoking status, DM, HTN, dyslipidemia, diagnosis duration, blood pressure category (controlled vs non-controlled), and HbA1c category (controlled vs non-controlled).

The Chi-square test was used to assess the distribution of anti-platelet and statin intensity among ARBs and the control group. Notably, 55% (n = 53) of the control group used aspirin + clopidogrel as an antiplatelet regimen compared to 61% (n = 22) of the ARBs group (P-value 0.267). Additionally, 42% (n = 15) of Patients in the ARBs group used high-intensity statins compared to 39% (n = 38) of the control group (P-value 0.130).

## Discussion

This study found that patients exposed to ARBs have a decreased frequency of revascularization procedures by 0.536 compared to those who are not exposed (P-value 0.025;95% CI −1.004 – - 0.068), after adjusting for confounding variables. Additionally, the likelihood of having advanced PAD stage (stages III and IV) is 71% lower in the ARB-exposed group than in the unexposed group (P-value = 0.034, odds ratio [OR]= 0.294, 95% CI = 0.095–0.911). Regarding limb-salvage outcomes, the odds of amputation in the ARB group are 79.2% lower than in the control group (P-value = 0.011, OR = 0.208, 95% CI = 0.062–0.700).

Patients in the ARBs group experienced lower rates of lower limb amputation compared to those in the control group. After adjusting for relevant covariates—including smoking status, DM, HTN, dyslipidemia, duration of diagnosis, blood

**Table 1. Baseline characteristics of study participants.**

| Characteristic | Control | ARBs | P value |
|---|---|---|---|
| Age (years) | 60.37 (10.665) | 63.06(9.184) | 0.448 |
| Duration of PAD (years) | 3.67 (2.37) | 3.358 (2.419) | 0.876 |
| SBP (mmHg) | 125.62 (15.38) | 138.86 (14.606) | 0.654 |
| HbA1c% | 7.195 (2.152) | 8.347 (2.485) | 0.342 |
| Male | 90 (92.8%) | 33 (91.7%) | 1.00 |
| Smoking status | | | 0.222 |
| Non-smoker | 11 (11.3%) | 4 (11.1%) | |
| Ex-smoker | 14 (14.4%) | 10 (27.8%) | |
| Smoker | 72 (74.2%) | 22 (61.1%) | |
| Diabetic | 67 (69.1%) | 27 (75.0%) | 0.505 |
| Has Hypertension | 43 (44.3%) | 32 (88.9%) | >0.001 |
| Has Dyslipidemia | 11 (11.3%) | 9 (25.0%) | 0.05 |
| Anti-platelets therapy | | | 0.267 |
| No anti-platelets | 8 (8.02%) | 0 (0%) | |
| ASA | 33 (34.0%) | 12 (33.3%) | |
| Clopidogrel | 3 (3.1%) | 2 (5.6%) | |
| Combination | 53 (54.6%) | 22 (61.1%) | |
| Statin use | | | 0.13 |
| No statin | 32 (33.0%) | 6 (16.7%) | |
| Moderate intensity | 27 (27.8%) | 15 (41.7%) | |
| High intensity | 38 (39.2%) | 15 (41.7%) | |
| Fontaine stage | | | 0.142 |
| Early stage (Stage IIa, IIb) | 67 (69%) | 30 (83.3%) | |
| Advanced stage (Stage III, IV) | 30 (31%) | 6 (16.7%) | |
| Controlled BP (I.e. BP<130) | 64 (66.0%) | 11 (30.6%) | >0.001 |
| Controlled HbA1c (I.e. HbA1c<7) | 45 (46.4%) | 9 (25.0%) | 0.026 |

Data are presented as means (SD) or numbers (%).

pressure category, and HbA1c category—ARBs were associated with reduced odds of amputation. This finding is consistent with the study by N. Elsayed et al., which demonstrated that the use of ARBs or ACEIs was linked to significantly improved amputation-free survival compared to controls (P-value<0.001) [13].

However, these results are inconsistent with the findings of E. J. Armstrong, who found that the use of ACEIs/ARBs was not significantly associated with major adverse limb events or major amputations [7]. This discrepancy may be explained by several factors: first, Armstrong's follow-up period was only three years after revascularization, and the sample consisted solely of patients with critical limb ischemia, defined by Rutherford category 4–6 disease (which includes ischemic rest pain, minor tissue loss, or major tissue loss). Additionally, Armstrong's study adjusted for different covariates, and the exposed group contained a higher proportion of diabetic and hypertensive patients (P-value <0.0050). S. Z. Khan et al. reached similar conclusions to Armstrong [8], and interestingly, both studies employed the same methodology.

Patients in the ARBs group showed a significantly lower frequency of revascularization interventions after adjusting for the aforementioned confounders. However, there was no significant association between the use of ARBs and the number of occluded arteries, suggesting that a larger sample size may be needed to detect such an association. This aligns with the results of a prospective randomized, single-blinded trial by W. Yao et al., which found that taking valsartan for six

months after stent placement was associated with a restenosis rate (defined as stenosis of 50% or more of the artery) of 11.22%, compared to 30.21% in the control group (P-value 0.001) [12].

The use of ARBs was the only independent variable that achieved significant levels in relation to more advanced disease stages, suggesting a potential protective effect against advanced disease stages. This finding was adjusted for factors such as smoking status, DM, HTN, dyslipidemia, duration of diagnosis, blood pressure category, and HbA1c category. Our study is the first to explore this association between the ARB group and the control group. In a study conducted by D. J. Margolis, the differences between ACEIs and ARBs were examined with regard to diabetic foot ulcers and limb amputations in diabetic patients. The research indicated that ARBs were associated with a lower incidence of diabetic foot ulcers (HR = 0.44, 95% CI 0.29, 0.65) and limb amputation (HR = 0.45, P-value 0.001, 95% CI 0.22–0.91) in patients with concurrent PAD compared to the ACEIs group. They found that ACEIs are more protective than ARBs for the first year of the study duration then the relationship got flipped over the next couple of years favoring ARBs, given the ACEI effect on bradykinin and inflammatory markers; it may delay or affect wound healing hypothetically [11]. Although these results support our findings regarding ARBs protective effect against progression to more advanced stages, this distinction between ARBs and ACEIs contradicts the more robustly designed published trials and simply could arise from patients' non-adherence, this set the stage for future research to confirm or contradict this ACEI versus ARBs distinction.

This study offers several important strengths, particularly in advancing understanding of ARB use in the treatment of PAD within the Jordanian context. Baseline characteristics were well balanced between groups, reducing the potential for selection bias. Despite limitations in sample recruitment, post hoc power analysis confirmed that the achieved sample size was sufficient to detect the observed effect size with adequate statistical power, thereby supporting the reliability of the findings..A follow-up period of approximately seven years allowed sufficient time for clinical outcomes to develop. Drawing participants from a tertiary hospital enabled the inclusion of a diverse patient population, thereby enhancing the generalizability of the findings to the northern region of Jordan. The clinical outcomes assessed are validated indicators of PAD progression and hold direct relevance for treatment decisions. Additionally, the completeness of the datasets—free from missing values—ensured rigorous statistical analysis. Notably, this is the first study of its kind in Jordan to explore the impact of ARBs, specifically, in the management of PAD, contributing valuable local evidence to guide future clinical practice.

This study presents several potential limitations. Firstly, the data were obtained from a single center, which may impact the generalizability of the findings. Nonetheless, because KAUH is a tertiary hospital that serves patients from across northern Jordan, this limitation may be somewhat mitigated. Secondly, the retrospective design inherently limits control over confounding factors, as evidenced by the model fitting results. While we adjusted for known confounders, the models did not achieve a satisfactory fit, likely due to the presence of unmeasured confounders and the relatively small sample size. Furthermore, there were notable differences in baseline characteristics between the groups. Although propensity score matching (PSM) is a widely used technique to address such imbalances, we chose not to apply it because of the limited sample size. Future research with a larger sample and a prospective design is necessary to validate these findings and to establish a stronger causal relationship between ARBs use and PAD outcomes.

The findings of this study provide further evidence that ARBs may offer potential benefits in the progression of PAD beyond merely lowering blood pressure. As a result, Canadian guidelines have incorporated ARBs alongside ACEIs as a first choice into the treatment algorithm for patients with both PAD and HTN [19]. However, additional research is necessary to explore this benefit more precisely, employing a larger sample size, prospective follow-up, and randomized sampling to establish causality regarding the effects of ARBs on disease progression. Our team is currently conducting a prospective cohort study aimed at investigating the association between ARBs and various clinical outcomes related to peripheral ischemia, including claudication, ankle-brachial index, and quality of life.

## Conclusion

PAD is a common condition worldwide that greatly impacts both mortality and morbidity rates. ARBs may serve as a promising protective treatment option, potentially benefiting disease progression by lowering the risk of amputation and reducing the need for revascularization procedures.

## Supporting information

**S1 File. Anonymized minimal dataset used for analysis in this study.**
(XLSX)

**S2 File. STROBE checklist for reporting observational studies.**
(DOCX)

## Acknowledgments

The authors would like to acknowledge the Deanship of Research at Jordan University of Science and Technology for approving this research (approval number: 2023/585).

## Author contributions

**Conceptualization:** Eilaf Mohammad Al -Rabab'ah, Rawand A. Khasawneh.

**Data curation:** Mahar Zaid Kasasbeh, Eilaf Mohammad Al -Rabab'ah.

**Formal analysis:** Mahar Zaid Kasasbeh, Eilaf Mohammad Al -Rabab'ah, Samah F. Al-Shatnawi.

**Investigation:** Mahar Zaid Kasasbeh, Eilaf Mohammad Al -Rabab'ah, Rawand A. Khasawneh, Fadi N. Asfar.

**Methodology:** Mahar Zaid Kasasbeh, Eilaf Mohammad Al -Rabab'ah, Rawand A. Khasawneh.

**Project administration:** Mahar Zaid Kasasbeh, Eilaf Mohammad Al -Rabab'ah, Rawand A. Khasawneh.

**Resources:** Mahar Zaid Kasasbeh, Eilaf Mohammad Al -Rabab'ah, Rawand A. Khasawneh.

**Software:** Mahar Zaid Kasasbeh, Eilaf Mohammad Al -Rabab'ah, Samah F. Al-Shatnawi.

**Supervision:** Rawand A. Khasawneh.

**Validation:** Mahar Zaid Kasasbeh, Eilaf Mohammad Al -Rabab'ah, Rawand A. Khasawneh, Samah F. Al-Shatnawi, Nour A. Al-Sawalha.

**Visualization:** Mahar Zaid Kasasbeh, Eilaf Mohammad Al -Rabab'ah, Rawand A. Khasawneh.

**Writing – original draft:** Mahar Zaid Kasasbeh, Eilaf Mohammad Al -Rabab'ah, Rawand A. Khasawneh, Fadi N. Asfar.

**Writing – review & editing:** Mahar Zaid Kasasbeh, Eilaf Mohammad Al -Rabab'ah, Rawand A. Khasawneh, Fadi N. Asfar, Samah F. Al-Shatnawi, Nour A. Al-Sawalha.

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
