## [Decision Letter · Decision Letter 0]

29 Jul 2025

Dear Dr. Kasasbeh,

We look forward to receiving your revised manuscript.

Kind regards,

Hean Teik Ong, FRCP, FACC

Academic Editor

PLOS ONE

Journal Requirements:

2 Thank you for stating the following in the Acknowledgments Section of your manuscript: [The authors would like to acknowledge the Deanship of Research at Jordan University of Science and Technology for approving this research (approval number: 825/2022).]

Please remove any funding-related text from the manuscript and let us know how you would like to update your Funding Statement. Currently, your Funding Statement reads as follows: “The authors received no specific funding for this work.”

3. Please upload a copy of Supporting Information Figure/Table/etc. S1 Fig. [Sampling flowchart]which you refer to in your text on page 14.

4. Please include captions for your Supporting Information files at the end of your manuscript, and update any in-text citations to match accordingly. Please see our Supporting Information guidelines for more information: http://journals.plos.org/plosone/s/supporting-information .

Additional Editor Comments:

Please make minor revision to the article to address reviewer comments.

Reviewers' comments:

Reviewer's Responses to Questions

**Comments to the Author**

1. Is the manuscript technically sound, and do the data support the conclusions?

Reviewer #1: Partly

Reviewer #2: Yes

2. Has the statistical analysis been performed appropriately and rigorously?

Reviewer #1: Yes

Reviewer #2: Yes

3. Have the authors made all data underlying the findings in their manuscript fully available?

Reviewer #1: Yes

Reviewer #2: Yes

4. Is the manuscript presented in an intelligible fashion and written in standard English?

Reviewer #1: Yes

Reviewer #2: Yes

Reviewer #1: 1. Consider citing methodological references or choosing more conventional sample size parameters.

2.The manuscript refers to 79.2% "lower limb amputation" without explicitly defining this as relative risk reduction. For clarity and transparency, consider rephrasing to “odds of limb amputation were 79.2% lower in the ARB group” and provide absolute event rates for both groups.

3. Provide a brief explanation of the Fontaine classification system in the methods for international readers.

4. The statement that ARBs were associated with "79.2% lower limb amputation" should be phrased more clearly.

Reviewer #2: A meaningful and well-executed study exploring the effect of ARBs on PAD progression, especially in Middle Eastern populations. The key strengths include a well-defined cohort, complete data extraction, and robust statistical adjustments. The findings provide evidence suggesting a protective role of ARBs in reducing PAD severity, need for revascularization, and amputation.

**Do you want your identity to be public for this peer review?** For information about this choice, including consent withdrawal, please see our Privacy Policy

Reviewer #1: **Yes: ** Tan Boon Seng

Reviewer #2: **Yes: ** Chong Hui Khaw

---

## [Author Response · Author response to Decision Letter 1]

31 Jul 2025

The effect of angiotensin receptor blockers on peripheral artery disease progression: A retrospective cohort study in Jordan

PONE-D-25-31876

PLOS ONE

Mahar Zaid Kasasbeh

July, 2025

Dear Editor-in-Chief,

I would like to express my sincere gratitude for considering our manuscript entitled: “The effect of angiotensin receptor blockers on peripheral artery disease progression: A retrospective cohort study in Jordan” for publication in your esteemed and scientifically respected journal. I kindly ask you to consider our revised version for further evaluation, taking into account that all comments and recommendations raised by your kind editorial office and the respected reviewer have been addressed thoroughly, as detailed in this response.

Please find below a point-by-point response to the editorial and reviewer comments. The comments are highlighted in bold, followed by our respective responses. Kindly notice that the reported lines in where the changes were made is based on the manuscript with track changes

2.Thank you for stating the following in the Acknowledgments Section of your manuscript: [The authors would like to acknowledge the Deanship of Research at Jordan University of Science and Technology for approving this research (approval number: 825/2022).] We note that you have provided funding information that is not currently declared in your Funding Statement. However, funding information should not appear in the Acknowledgments section or other areas of your manuscript. We will only publish funding information present in the Funding Statement section of the online submission form.

Thank you for your feedback. We appreciate your concern regarding the potential confusion between our funding statement and the acknowledgment section. We would like to clarify that our research did not receive any funding. The acknowledgment of the Deanship of Research at our university was included solely to recognize their support in facilitating IRB approval and providing access to the hospital database.

All references have been reviewed for retraction or corrections using the Retraction Watch database and PubMed. As of July, 31th, 2025, none of the cited works have been retracted or corrected in a way that affects the study’s content

Additional Editor Comments to Author:

Reviewer #1 Comments to Author:

1. Consider citing methodological references or choosing more conventional sample size parameters.

Thank you for the insightful comment. We have revised the sample size calculation section accordingly. Although we were unable to locate a recent and reliable reference for the equation previously used to estimate the odds ratio (OR) from the hazard ratio (HR), we have now provided a detailed explanation of the rationale behind our approach to the sample size calculation, along with its implications. Please refer to the revised sections in the Methods (lines 133–149) and Discussion (lines 231–233).

2.The manuscript refers to 79.2% "lower limb amputation" without explicitly defining this as relative risk reduction. For clarity and transparency, consider rephrasing to “odds of limb amputation were 79.2% lower in the ARB group” and provide absolute event rates for both groups.

We appreciate the suggestion. We have revised the sentence reporting the percentage of amputations in each group and have explicitly included the term absolute amputation rate to enhance clarity. Please see the updated text in the Results section (line 167).

3. Provide a brief explanation of the Fontaine classification system in the methods for international readers.

A brief explanation of the Fontaine classification system was already provided in the Methods section. Please refer to the Methods section (lines 110–113).

4. The statement that ARBs were associated with "79.2% lower limb amputation" should be phrased more clearly.

Thank you for the insightful comment. We appreciate your detailed recommendations. In response, we have revised the statement to read: “The odds of amputation in the ARBs group were 79.2% lower than in the control group.”

At the end, I would like to extend our sincere thanks for allowing us the opportunity to revise and resubmit our manuscript. We believe that the changes made in response to your kind guidance have substantially improved the manuscript, and we respectfully request your reconsideration for publication.

Sincerely,

---

## [Editor Report · Decision Letter 1]

4 Aug 2025

The effect of angiotensin receptor blockers on peripheral artery disease progression: A retrospective cohort study in Jordan

PONE-D-25-31876R1

Dear Dr. Mahar Zaid Kasasbeh,

We’re pleased to inform you that your manuscript has been judged scientifically suitable for publication and will be formally accepted for publication once it meets all outstanding technical requirements.

Kind regards,

Hean Teik Ong, FRCP, FACC

Academic Editor

PLOS ONE
---

## [Editor Report · Acceptance letter]

PONE-D-25-31876R1

PLOS ONE

Dear Dr. Kasasbeh,

I'm pleased to inform you that your manuscript has been deemed suitable for publication in PLOS ONE. Congratulations! Your manuscript is now being handed over to our production team.

Kind regards,

on behalf of

Dr. Hean Teik Ong

Academic Editor

PLOS ONE